# Breaking Barriers: Candidalysin Disrupts Epithelial Integrity and Induces Inflammation in a Gut-on-Chip Model

**DOI:** 10.3390/toxins17020089

**Published:** 2025-02-14

**Authors:** Moran Morelli, Karla Queiroz

**Affiliations:** MIMETAS B.V., De Limes 7, 2342 DH Oegstgeest, The Netherlands

**Keywords:** gut-on-chip, organ-on-a-chip, microphysiological systems, host-microbial interactions, candidalysin, barrier function, toxicology screening, *Candida albicans*, organoids

## Abstract

*Candida albicans* is an opportunistic pathogenic yeast commonly found in the gastrointestinal tract of healthy humans. Under certain conditions, it can become invasive and cause life-threatening systemic infections. One mechanism used by *C.albicans* to breach the epithelial barrier is the secretion of candidalysin, a cytolytic peptide toxin. Candidalysin damages epithelial membranes and activates the innate immune response, making it key to *C.albicans*’ pathogenicity and a promising therapeutic target. Although candidalysin mediates *C. albicans* translocation through intestinal layers, its impact on epithelial responses is not fully understood. This study aims to characterize this response and develop scalable, quantitative methodologies to assess candidalysin’s toxicological effects using gut-on-chip models. We used the OrganoPlate^®^ platform to expose Caco-2 tubules to candidalysin and evaluated their response with trans-epithelial electrical resistance (TEER), protein detection, and immunostaining. We then validated our findings in a proof-of-concept experiment using human intestinal organoid tubules. Candidalysin impaired barrier integrity, induced actin remodeling, and increased cell permeability. It also induced the release of LDH, cytokines, and the antimicrobial peptide LL37, suggesting cellular damage, inflammation, and antimicrobial activity. This study strengthens our understanding of candidalysin’s role in *C. albicans* pathogenesis and suggests new therapeutic strategies targeting this toxin. Moreover, patient-derived organoids show promise for capturing patient heterogeneity and developing personalized treatments.

## 1. Introduction

The intestinal mucosa is the first line of defense against microbial invasion. Intestinal epithelial cells (IECs) form a barrier, which separates the sterile host environment from the gut microbiota and the external environment. Perturbations of the host–microbial interplay such as an imbalance in the microbial community, a breach in the barrier, or an impairment of the host’s immune system can lead to disease [1].

Candida species are among the leading fungal pathogens, significantly contributing to morbidity and mortality [2]. Among these, *C. albicans* is the most prevalent cause of life-threatening systemic candidiasis and is a major contributor to nosocomial infections, particularly in Intensive Care Units (ICUs) [2,3]. *C. albicans* is an opportunistic pathogen that frequently inhabits the mucosal surfaces of humans, with its main reservoir being the gastrointestinal tract [4,5,6,7]. A compromised immune system and the use of broad-spectrum antibiotics can induce the transition of *C. albicans* from commensalism to pathogenicity, where it can breach the intestinal barrier and reach the bloodstream to cause systemic candidiasis [8,9,10]. One of the key mechanisms mediating the translocation of *C. albicans* through the intestinal layers is the release of the virulence factor candidalysin [11,12].

Candidalysin is a cytolytic peptide playing a dual role in *C. albicans* infection. On one hand, it acts as a virulence factor by directly damaging the host’s epithelium through the formation of membrane pores [11,13,14]. On the other hand, candidalysin acts as an immunomodulatory molecule recognized by the host to initiate an immune response [11,15,16]. Characterizing the epithelial response to candidalysin is therefore of critical importance to better manage candidiasis.

So far, most of the in vitro research exploring the mechanisms of candidalysin has been conducted on oral epithelial cell lines, where candidalysin has been shown to trigger the release of cytokines, chemokines, alarmins, and antimicrobial peptides [16,17,18,19]; to induce epithelial stress such as mitochondrial dysfunction, ATP depletion, and epithelial necrosis [20]; and to trigger Ca^2+^ influx and breakdown of F-actin [21]. Although it is known that the main reservoir of *C. albicans* is the gut [4,22], few studies explored the effect of candidalysin on intestinal epithelial cells (IECs).

Allert et al. demonstrated that a *C. albicans* mutant lacking candidalysin showed reduced barrier disruption and translocation in the Caco-2 brush border subclone C2BBe1, indicating the toxin’s essential role in fungal penetration through the intestinal wall [10]. A follow-up study found that intestinal translocation was promoted by the acquisition of host–cell zinc by *C. albicans*, while at the same time, host NFκB signaling was protecting epithelial integrity, mitigating candidalysin-induced damage and limiting fungal translocation [23]. While previous studies have explored the role of candidalysin in epithelial damage and translocation, they primarily used static models or non-gut epithelial cells, which are limited in their ability to dynamically assess barrier function and immune responses within a physiologically relevant gut environment. Moreover, these studies have largely focused on understanding fundamental mechanisms, with less emphasis on developing scalable platforms suitable for therapeutic applications.

Here, we used a membrane-free gut-on-chip model (Figure 1) to evaluate the intestinal response to candidalysin in the cell line Caco-2, using scalable and quantitative readouts. We characterized the response to the toxin by studying the barrier integrity, morphology, permeability, and release of epithelial mediators at the luminal side of the model. To validate our findings in a more relevant gut-on-chip model, we included a proof-of-concept experiment using patient-derived colon organoids to study barrier integrity, morphology, and permeability after candidalysin exposure.

In summary, our study presents a novel gut-on-chip platform with robust, scalable readouts to evaluate candidalysin’s effects on intestinal epithelial cells. This system is ideal for phenotypic screenings to identify therapeutic molecules and can evolve to include other relevant cell types, enabling comprehensive models for both toxicological assessment and personalized drug development targeting microbial virulence factors.

## 2. Results

### 2.1. Candidalysin Impairs Barrier Integrity of Caco-2 Tubules and Increases Their Permeability

To determine candidalysin-mediated effects on the intestinal barrier integrity, we used the OrganoTEER^®^ (Figure 2a,b) to measure TEER of the Caco-2 tubules after 30 min, 60 min, and 120 min candidalysin exposure (37 and 75 µM). Both concentrations significantly decreased TEER after all timepoints compared to the vehicle (Figure 2c,d). When comparing the effect over time, although 37 µM candidalysin exposure showed a trend of recovering TEER over time, and 75 µM candidalysin showed a trend of decreasing TEER over time, these changes were not statistically significant (*p* > 0.05) (Figure 2c,d).

We next investigated the permeability of the Caco-2 tubules after 37 and 75 µM candidalysin exposure, using fluorescent molecules of two different sizes. Figure 3a–c illustrates the experimental setup and cross-sectional views of the tubules, with the top illustration representing an intact barrier and the bottom depicting a compromised barrier with increased permeability. Permeability to sodium fluorescein (0.4 kDa) was significantly increased after 30, 60, and 120 min incubation with 37 µM candidalysin (Figure 3d), while permeability to dextran 4.4 kDa was significantly increased only after 30 min 37 µM candidalysin exposure (Figure 3f). Briefly, 75 uM candidalysin increased permeability to sodium fluorescein after all incubation times (Figure 3e), and permeability to 4.4 kDa dextran after 30 and 120 min (Figure 3e). Following the evaluation of permeability across various candidalysin exposure conditions, we selected the 120 min incubation with 75 µM candidalysin for further analysis.

We then assessed whether the expression of the permeability marker DRAQ7^TM^ was increased after 75 µM candidalysin exposure for 120 min when compared to the vehicle. Figure 4a shows images of representative chips exposed to vehicle or candidalysin and stained for DRAQ7 and Hoechst. Permeable cells’ nuclei and cell count (total nuclei) quantification are shown in the middle and bottom rows, respectively. Briefly, 75 µM candidalysin significantly increased the number of DRAQ7 positive cells compared to vehicle (Figure 4b).

### 2.2. Candidalysin Induces Actin Remodeling in Caco-2 Tubules

Next, we investigated the morphology of the tubules after candidalysin exposure. Cells were exposed to vehicle or 75 µM candidalysin for 120 min. Figure 5a shows images of representative chips exposed to vehicle or candidalysin and stained for actin and nuclei. Actin-positive objects and cell count (nuclei) quantification are shown in the middle and bottom rows, respectively. Briefly, 75 µM candidalysin significantly increased the number of actin-positive objects (Figure 5b), the total actin area (Figure 5c), and the perimeter of actin objects (Figure 5d) compared to the vehicle. All parameters were normalized to the total cell number.

### 2.3. Exposure to Candidalysin Induces Cytotoxicity and Secretion of Inflammatory Markers in Caco-2 Tubules

To evaluate the effect of candidalysin on the cellular activation of Caco-2 tubules, the production of epithelial inflammatory mediators LL37, S100A8/A9, MCP-1, IP-10, IL-8, IL-6, IL-1beta, IL-1alpha, GM-CSF, MIP-3 alpha, and G-CSF was quantified using ELISA and Luminex technology. Caco-2 cells produced no or low amounts of these analytes in non-triggered conditions. After 120 min 75 µM candidalysin, the secretion of all mediators increased significantly (Figure 6), except S100A8/A9, which was not detected. We used the lactate dehydrogenase (LDH) assay to measure cytotoxicity after candidalysin exposure. LDH was significantly higher in chips treated with 120 min 75 µM candidalysin compared to vehicle (Figure 6).

### 2.4. Assessment of Response to Candidalysin in Human Colon Organoids

To validate our results obtained from Caco-2 tubules and to explore future applications in a patient-derived model, we conducted a proof-of-concept experiment using human colon organoid tubules. We used the OrganoTEER to measure the barrier integrity of colon organoids following a 120 min exposure to 0, 18, 37, and 75 µM candidalysin. Subsequently, we fixed the plates and performed DRAQ7 and actin staining (Figure 7a) and quantification. While the pre-exposure washing step led to a reduction in TEER across all conditions, trigger-specific effects were still observed, with 75 µM candidalysin significantly decreasing TEER after 120 min exposure (Figure 7b). Both 37 and 75 µM candidalysin increased permeability to DRAQ7 compared to the vehicle (Figure 7c). Furthermore, 75 µM candidalysin led to significant actin remodeling, as evidenced by increases in the number (Figure 7d), total area (Figure 7e), and perimeter of actin objects (Figure 7f).

## 3. Discussion

This study aimed to evaluate the phenotypic response of IECs to candidalysin, focusing on barrier integrity and inflammation. Given that candidalysin forms pores and damages the epithelium [13,14], we hypothesized it would impair the barrier integrity and increase the permeability in our model. We also expected to detect toxicity and inflammation mediators, as described in studies using oral epithelial cells [17,19].

To test these hypotheses, we used the OrganoPlate technology, allowing the culture of up to 64 perfused intestinal Caco-2 tubules in a membrane-free system [24]. This platform allowed the rapid assessment of the barrier integrity of the gut tubules [25], along with easy imaging and sampling of luminal medium to detect inflammatory markers. We found that candidalysin impaired the barrier integrity of Caco-2 tubules (measured by TEER and barrier integrity assay), increased permeability (measured by DRAQ7 staining), and induced actin remodeling and the secretion of inflammatory mediators. We then validated our results in a more complex model using patient-derived organoid tubules exposed to candidalysin, measuring TEER, DRAQ7 permeability, and actin remodeling.

The disruption of barrier integrity is a hallmark of mucosal inflammation. Using a combination of readouts, we observed that candidalysin impaired barrier integrity and increased permeability. In Caco-2 tubules, only TEER detected significant differences at all timepoints after 37 µM candidalysin exposure, suggesting that TEER is more sensitive than a barrier integrity assay [24,25]. In colon organoid tubules, TEER was less sensitive than in Caco-2, with only 75 µM of candidalysin showing a significant decrease after 120 min exposure. This reduced sensitivity may be due to the increased complexity of colon organoids compared to Caco-2 and could be influenced by the washing step before exposure, as indicated by decreased TEER in the vehicle control.

While several studies reported loss of barrier integrity after infection with *C. albicans* in IECs [7,10,26] and other epithelial cell types [27,28], to date, only one study used exogenous candidalysin in IECs. Contrary to our observations, they did not observe a decrease in TEER or an increase in 4 kDa dextran permeability after 24 h of 70 µM candidalysin exposure using the Caco-2 subclone C2BBe1 in Transwells [10]. This discrepancy likely arises from model differences (dynamic gut-on-chip vs. static Transwells), exposure duration (acute 2 h vs. prolonged 24 h), cell line sensitivity, and measurement sensitivity (more responsive TEER in the gut-on-chip [25]). These differences highlight how model choices impact findings, suggesting the gut-on-chip system could better capture acute barrier disruption, influencing interpretations of candidalysin’s role in epithelial damage.

The actin cytoskeleton mediates the disruption of mucosal barriers in inflamed tissues [29]. We therefore looked at the actin network after candidalysin exposure and observed remodeling in Caco-2 and organoids, confirming results observed in oral epithelial cells [21]. Notably, in colon organoids, actin clumps were found to co-localize with DRAQ7-positive nuclei, suggesting a localized effect of candidalysin in these areas.

Similarly to observations in other cell types [12,17,18,19], we observed the induction of several candidalysin-induced proteins in Caco-2. Along with the release of LL-37, G-CSF, GM-CSF, IL-6, IL-1a, IL-1b, MCP-1, and IL-8, we also observed the candidalysin-induced release of IP-10 and MIP-3 in our cultures. Notably, S100A8/A9 release was not induced by candidalysin in our system, in line with findings in TR146 cells [17] and the fact that these molecules are mainly released by neutrophils and macrophages [30]. The secretion of LL-37, an antimicrobial peptide, is induced by candidalysin, highlighting its role in the host defense mechanism. However, further experiments with additional stimuli are needed to determine the specificity of this response.

Our model has inherent trade-offs in its design choices. The use of exogenous candidalysin provides precise control over epithelial responses and early infection mechanisms but does not replicate the delivery of candidalysin to host cell membranes [31]. We cannot address direct fungal–host interactions, such as the use of host zinc by the yeast to promote translocation and virulence [23] or interactions between *C. albicans* and the rest of the microbiota, where it was shown that candidalysin can inhibit bacterial species [32]. The current system also operates without immune components, although other gut-on-chip models have incorporated both living yeast and immune cells [33]. While some of these more complex models can recapitulate immune migration [34,35], they are challenging to scale up for therapeutic screening [36]. These design choices prioritize reproducibility and scalability, making our platform particularly suited for studying epithelial responses to candidalysin.

While we successfully demonstrated candidalysin’s effects on barrier function and morphology in organoid tubules, protein detection was not included in this initial proof-of-concept study. Future work using organoids would provide valuable insights into protein expression responses in a more physiologically relevant environment. Additionally, testing multiple donors would help capture the variability in responses to candidalysin.

While organoids are a promising model for studying intestinal barrier function, they still present technical limitations that affect direct comparisons with traditional models like Caco-2. Different experimental procedures were required for each model, leading to baseline TEER changes in organoids. These methodological differences mean that results from Caco-2 and organoid tubules cannot be directly compared, highlighting a broader challenge in the field—the need for standardized culture conditions and experimental protocols across different model systems.

In summary, our gut-on-chip model effectively evaluated the phenotypic response to candidalysin in intestinal epithelial cells, providing insights into the toxin’s mechanisms. This system’s ability to multiplex readouts makes it valuable for screening therapeutic compounds, including candidalysin-neutralizing nanobodies [37]. The platform can be adapted to study candidalysins from non-albicans Candida species [19] and other toxins [38]. Moreover, the integration of patient-derived organoids could enable personalized modeling for patients with candidiasis or other gastrointestinal disorders. Beyond fungal infections, this platform provides a robust tool for investigating microbial metabolites and screening therapeutic compounds that target epithelial barrier function.

## 4. Materials and Methods

### 4.1. Cell Culture

A detailed list of all reagents, including catalog numbers and sources, is provided in the Appendix A.

#### 4.1.1. OrganoReady Colon Caco-2

OrganoReady^®^ Colon Caco-2 3-lane 40 (MI-OR-CC-01, MIMETAS, Oegstgeest, The Netherlands) was cultured according to the manufacturer’s instructions (Figure 1). Medium was replaced with Caco-2 medium on the day of receiving. OrganoReady Colon Caco-2 3-lane 40 are ready-to-use Caco-2 tubes in OrganoPlate that follow a similar process to what was described by Trietsch and colleagues [24]. Perfusion flow was maintained by placing the plate on an OrganoFlow^®^ rocker (MI-OFPR-L MIMETAS, Oegstgeest, The Netherlands) set at 7 degrees with 8 min intervals optimized for the 3-lane 40. On the second day after receiving (day 6 after seeding), the Caco-2 medium was refreshed. Exposures were performed on day 4 after receiving (day 8 after seeding).

#### 4.1.2. OrganoReady Colon Organoid

OrganoReady^®^ Colon Organoid 3-lane 64 (MI-OR-CORG-02, MIMETAS, Oegstgeest, The Netherlands) was used to study the effect of candidalysin in colon organoids. Each plate contained 64 chips with colon organoid tubules, cultured using the following media: OrganoMedium Colon Organoid-ARM (Apical Recovery Medium), BRM (Basolateral Recovery Medium), ACM (Apical Culture Medium), and BCM (Basolateral Culture Medium), per the manufacturer’s instructions. Perfusion flow was maintained on an OrganoFlow rocker (MI-OFPR-L, MIMETAS, Oegstgeest, The Netherlands) set at 14 degrees with 8 min intervals optimized for the 3-lane 64. The tubules were derived from the healthy portion of a sigmoid biopsy from a 58-year-old Caucasian female with colorectal cancer.

### 4.2. Candidalysin Exposure

Candidalysin (SIIGIIMGILGNIPQVIQIIMSIVKAFKGNK) was synthesized by Peptide Protein Research Ltd. (Fareham, UK) and reconstituted in sterile purified water to 5 mg/mL (1.5 mM) for storage at −20 °C, prior to further dilution for individual experiments. The 37 µM and 75 µM concentrations were selected from the higher end of the range used in previous studies [12,17,20]. For organoid experiments, an additional 18 µM concentration was included to account for their potentially higher sensitivity as a primary cell model.

For exposures in Caco-2 tubules, medium was replaced with serum-free medium in all channels 24 h prior to exposure. Then, the medium of the apical channel was replaced with a serum-free medium containing candidalysin at the specified concentration.

Due to their increased sensitivity and the limitation of modifying their culture medium, colon organoid tubules were not serum-starved. Instead, all channels were washed with HBSS (Hank’s Balanced Salt Solution) prior to adding HBSS containing the specified concentration of candidalysin in the apical channel.

### 4.3. Protein Detection

Supernatants were collected from the apical channel and centrifuged for 10 min at 1500 rpm prior to aliquoting and storage at −80 °C.

LL37 ELISA kits were purchased from Elabscience (Houston, TX, USA) and used according to the manufacturer’s instructions. Absorbance was measured using a Spark Cyto plate reader (TECAN, Männedorf, Switzerland).

G-CSF, GM-CSF, IL-1 alpha, IL-1 beta, IL-6, IL-8, IP-10, MCP-1 (CCL2), MIP-3 alpha (CCL20), and S100A8/A9 were measured using a Luminex MAGPIX instrument (Thermo, Waltham, MA, USA) and a custom Procartaplex assay kit (Thermo, Waltham, MA, USA), following the manufacturer’s instructions. Plates were measured using a Luminex MAGPIX^®^ instrument and the Luminex xPONENT^®^ software (version 4.3). ProcartaPlex Analysis App was used to determine analyte concentrations.

We applied a single imputation by replacing values below the limit of quantification (LOQ) with LOQ/2 for statistical analysis. LOQ values, as provided by the manufacturer, are reported in the Table of reagents in the Appendix A.

### 4.4. Trans-Epithelial-Electrical-Resistance (TEER)

Trans-epithelial electrical resistance (TEER) was measured using an automated multichannel impedance spectrometer (OrganoTEER^®^, Figure 2a–c) to evaluate the integrity of the gut barrier [25]. An electrode board, designed to fit the 3-lane OrganoPlate was sterilized with 70% ethanol at least 1 h before measurement. The OrganoPlate was taken out of the incubator and equilibrated at room temperature for 30 min prior to the measurement to eliminate any temperature or flow effect. A baseline measurement was performed right after equilibration, and then the TEER was further measured at the indicated timepoints after candidalysin exposure.

For Caco-2, the OrganoTEER was set on “high TEER” and we used a TEER threshold of 350 Ω/cm^2^ at baseline, meaning that all tubules below 350 Ω/cm^2^ before exposure were excluded from the analysis. For colon organoids the OrganoTEER was set on “medium TEER” and we used a TEER threshold of 100 Ω/cm^2^ at baseline, meaning that all tubules below 100 Ω/cm^2^ before exposure were excluded from the analysis.

### 4.5. Barrier Integrity Assay

To further investigate the barrier integrity, the leakage of fluorescent molecules from the lumen to the ECM compartment was evaluated after exposure (Figure 3a–c). First, the medium from all inlets and outlets was removed. Then, 25 µL serum-free medium was added to the inlet and outlets of the middle and basal channels. Next, a dye solution consisting of Sodium fluorescein (0.4 kDa, 10 μg/mL, Sigma-Aldrich, St. Louis, MO, USA) and TRITC-dextran (4.4 kDa, 500 μg/mL, Sigma-Aldrich, USA) in serum-free medium was added to the inlet and outlet of the apical channel (40 and 30 µL, respectively). Then, the OrganoPlate was placed in an ImageXpress XLS Micro (Molecular Devices, San Jose, CA, USA) and imaged every 2 min for 7 timepoints. Leakage from the lumen into the ECM compartment was quantified according to the protocol described by Soragni et al. [39].

### 4.6. Cell Damage (LDH) Assay

Lactate dehydrogenase (LDH) activity was measured using the LDH-Glo™ Cytotoxicity Assay kit (Promega, Madison, WI, USA), following the manufacturer’s instructions. Supernatants were collected from the lumen and centrifuged for 10 min at 1500 rpm, prior to dilution in LDH assay buffer (1:100) and storage at −80 °C. Luminescence was measured in 96 W half area white microplates (Greiner bio-one, Kremsmünster, Austria) using a Spark Cyto plate reader (TECAN, Männedorf, Switzerland).

### 4.7. Immunofluorescence

For the visual characterization of the effect of candidalysin on actin cytoskeleton and cell permeability, tubules were directly fixed and stained on the OrganoPlate based on the protocol described by Trietsch et al. [24].

#### 4.7.1. DRAQ7 Staining

Before fixation, the medium in the tubule inlets and outlets was replaced by 20 μL of DRAQ7 dye (Biostatus, DR71000) diluted 1:100 in serum-free EMEM (Caco-2) or HBSS (colon organoids), and the cells were incubated during 30 min under continuous perfusion inside the incubator. OrganoPlates were then fixed and stained following the previously explained protocol.

#### 4.7.2. Fixation

Intestinal tubules were fixed with 3.7% formaldehyde (Sigma, 252,549) in HBSS with Calcium and Magnesium (Thermo Scientific, 14,025,092) for 15 min, washed twice with phosphate-buffered saline (PBS; Gibco, 70,013,065) for 5 min and then stored with 50 μL PBS per well at 4 °C until further staining.

#### 4.7.3. Actin and Nuclear Staining

Intestinal tubule cells were permeabilized with 0.03% Triton X-100 (Sigma, T8787) in PBS for 10 min and washed twice with 4% FBS in PBS solution. Actin and nuclear stainings were performed using the direct stains ActinGreen^TM^ 488 ReadyProbes^TM^ Reagent (Invitrogen, Waltham, MA, USA, R37110) and NucBlue^TM^ Fixed Cell ReadyProbes Reagent (Hoechst 33342, Invitrogen, R37605). Direct staining was performed following the manufacturer’s instructions and under constant flow.

For actin and Hoechst staining, two drops/mL were added in PBS, and 20 µL of the staining solution was added to the tube inlet and outlet (20 µL each) and incubated for 30 min at room temperature under continuous perfusion. The plates were then washed twice for 5 min with PBS and the bottom of the tubules was imaged as max projection using an ImageXpress Micro Confocal microscope (Molecular Devices, San Jose, CA, USA).

### 4.8. Intestinal Tubules Visualization and Imaging

Tubules were imaged using the ImageXpress^®^ Micro XLS (Molecular Devices, San Jose, CA, USA) and Micro XLS-C High Content Imaging Systems (Molecular Devices, San Jose, CA, USA) and processed using Fiji34 (Version 1.52d) to enhance contrast and improve visualization. To monitor the integrity of the tubules, phase-contrast images were recorded before and after exposure to candidalysin. This was routinely performed immediately after the measurement of the baseline TEER and after the required exposure time. Fixed and stained OrganoPlates were stored at 4 °C until imaging and equilibrated at room temperature at least 30 min before imaging. Maximum intensity projection images were saved as TIFF files after the confocal imaging of the stained cells.

### 4.9. Staining Quantification

Open-source cell image analysis software CellProfiler^TM^ (version 4.2.5) was used to process visual immunofluorescence images. An image analysis pipeline was designed to quantify the number and area of actin clumps and the number of cells, by the segmentation of the actin clumps and nuclei and the measurement of the total actin clump area. This pipeline was used to process the TIFF files that captured actin staining or nuclei staining, correspondingly. Parallelly, another pipeline was designed to process the TIFF files capturing DRAQ7 and nuclei staining, which allowed the segmentation of DRAQ7-positive cells and nuclei, to quantify the percentage of DRAQ7-positive (DRAQ7+) cells.

### 4.10. Statistics

Data analysis and visualization were conducted using R version 4.2.2 and RStudio version 2023.03.0-386.

All experiments were performed three times in an independent manner (N = 3), and the number of technical replicates per concentration (n) is indicated in the caption of each figure. The normality of The data distribution and homogeneity of variances were assessed using the Shapiro–Wilk test and Levene’s test, respectively. Differences among the two groups were compared using an independent T-test or Wilcoxon Rank-Sum Test (Mann–Whitney U Test). Differences among three groups or more were compared using one-way ANOVA followed by post hoc Tukey’s HSD, or Kruskal–Wallis tests followed by Dunn’s test, to obtain pairwise comparisons between the effect of exposure to the various candidalysin concentrations and exposure. *p* values were adjusted for multiple comparisons using the Bonferroni method. Adjusted *p* values of 5% or lower were considered statistically significant and were reported with asterisks: * *p* ≤ 0.05; ** *p* ≤ 0.01; *** *p* ≤ 0.001, **** *p* ≤ 0.0001. *p* values above 5% were considered not statistically significant (ns) and were not reported.

## Figures and Tables

**Figure 1 toxins-17-00089-f001:**
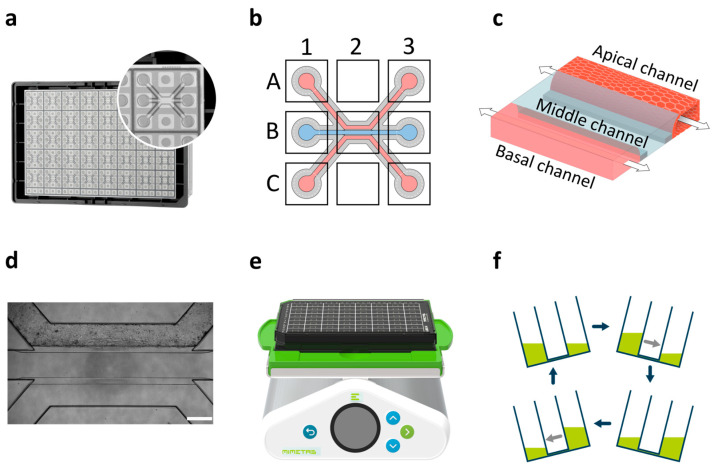
OrganoReady^®^ Caco-2 model. (**a**) Picture of an OrganoPlate^®^ 3-lane 40 with one microfluidic chip enlarged. (**b**) Illustration of a microfluidic chip. Apical channel can be accessed through A1 and A3. Middle channel can be accessed through B1 and B3. Basal channel can be accessed through C1 and C3. (**c**) Schematic of a Caco-2 tubule (apical channel) against a collagen ECM (middle channel). Basal channel contains culture medium. Arrows indicate flow direction. (**d**) Phase-contrast image of a Caco-2 tubule in the OrganoPlate 3-lane 40, scalebar is 100 µm. (**e**) OrganoPlate on an OrganoFlow^®^ rocker. The rocker angle is set at 7 degrees and oscillates every 8 min. (**f**) Illustration of bi-directional flow in the OrganoPlate.

**Figure 2 toxins-17-00089-f002:**
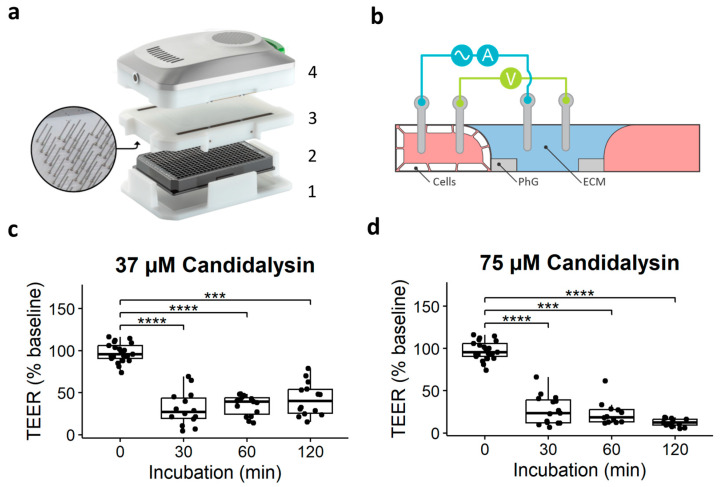
TEER measurements of Caco-2 tubules exposed to candidalysin. (**a**) Photograph of the OrganoTEER setup: the plate holder (1) holds the OrganoPlate (2) in which the electrode board (3) is positioned and connected to the measuring device (4). (**b**) Illustration of a cross-section of a microfluidic chip and positioning of the electrodes. PhG: phaseguide technology, allowing membrane-free separation of the channels. ECM: extracellular matrix. (**c**) TEER measurements after 30 min, 60 min, and 120 min exposure to 37 µM candidalysin, and (**d**) 75 µM candidalysin. Results are expressed as boxplots with individual measurements for each chip. Each chip is normalized to its baseline value before exposure. Data were analyzed using Kruskal–Wallis test with Dunn’s post hoc test (*** *p* ≤ 0.001, **** *p* ≤ 0.0001). Values were obtained from three independent experiments, with 3 to 8 chips per condition (N = 3, n = 3–8).

**Figure 3 toxins-17-00089-f003:**
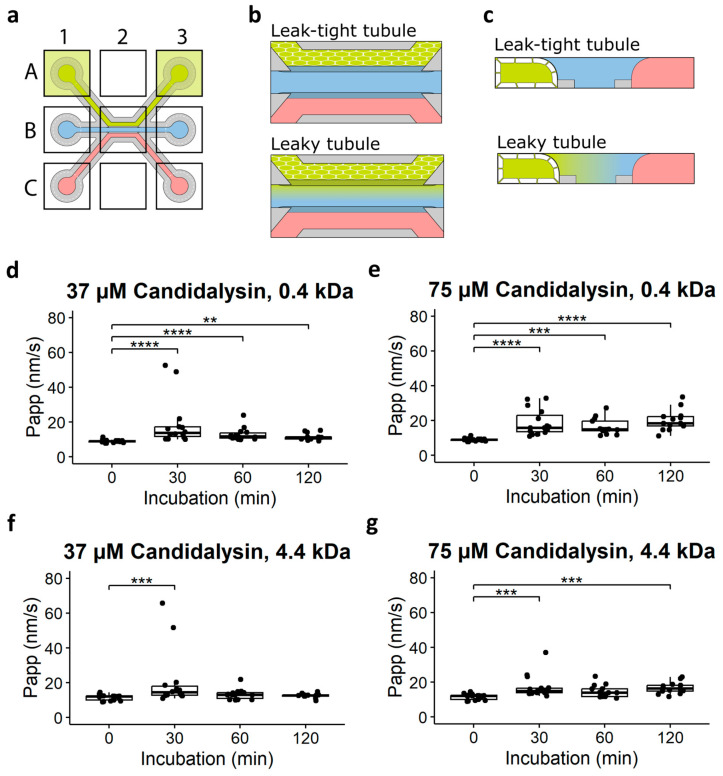
Caco-2 tubule permeability to fluorescent dyes after candidalysin exposure. (**a**) Illustration of a chip with the dye added to the lumen of the Caco-2 tubule. (**b**) Enlarged view of a chip with dye perfused in the lumen of the tubule. Top illustration shows a leak-tight tubule; bottom illustration shows a leaky tubule. (**c**) Cross-section of a chip to show a leak-tight tubule at the top and a leaky tubule at the bottom. (**d**) Permeability of Caco-2 tubule to 0.4 kDa dye after 30, 60, and 120 min exposure to 37 µM, and (**e**) 75 µM candidalysin. (**f**) Permeability of Caco-2 tubule to 4.4 kDa dye after 30, 60, and 120 min exposure to 37 µM, and (**g**) 75 µM candidalysin. (Results are expressed as boxplots with individual measurements for each chip. Data were analyzed using Kruskal–Wallis test with Dunn’s post hoc test (** *p* ≤ 0.01; *** *p* ≤ 0.001, **** *p* ≤ 0.0001). Values were obtained from three independent experiments, with 3 to 7 chips per condition (N = 3, n = 3–7).

**Figure 4 toxins-17-00089-f004:**
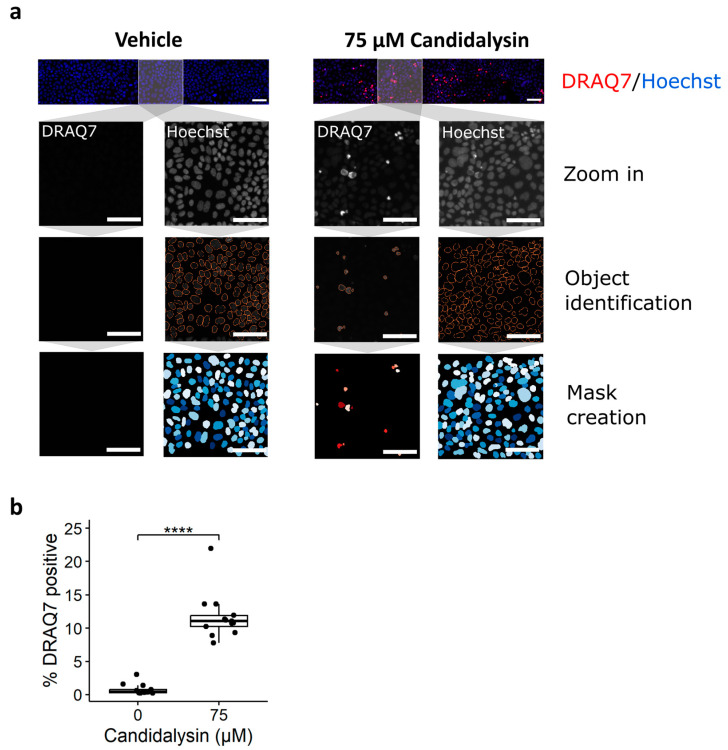
Candidalysin increases permeability of IECs. Caco-2 tubules exposed to candidalysin for 120 min were stained with DRAQ7 nuclear dye, which only penetrates permeabilized or dead cells, and with Hoechst. (**a**) A CellProfiler pipeline was designed to identify and quantify DRAQ7-positive and Hoechst-positive nuclei from confocal images. DRAQ7+ objects were filtered to only consider those overlapping with Hoechst positive objects. Pictures are representative of 3 independent experiments (N = 3). Scalebar is 100 µm. (**b**) The number of DRAQ7+ nuclei was normalized against the total number of nuclei (Hoechst positive) to give a percentage of DRAQ7 positive cells.) Results are expressed as boxplots with individual measurements for each chip. Data were analyzed using Wilcoxon Rank-Sum Test (**** *p* ≤ 0.0001). Values were obtained from three independent experiments, with 4 to 5 chips per condition (N = 3, n = 4–5).

**Figure 5 toxins-17-00089-f005:**
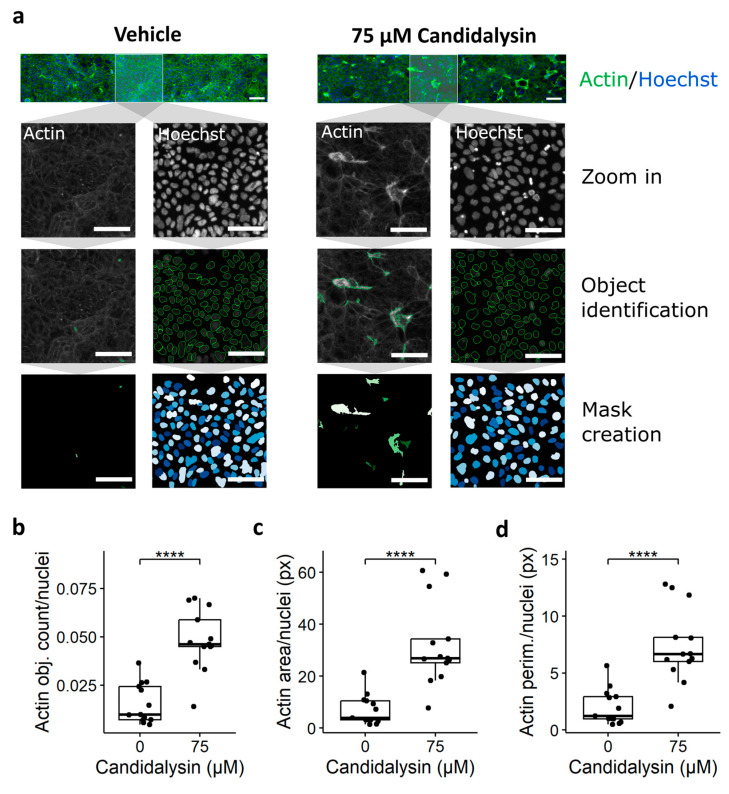
Candidalysin induces actin remodeling. Caco-2 tubules exposed to candidalysin for 120 min were stained with actin and Hoechst nuclear dye. (**a**) A CellProfiler pipeline was designed to identify and quantify actin-positive and Hoechst-positive objects from confocal images. Pictures are representative of 3 independent experiments (N = 3). Scalebar is 100 µm. The number (**b**), total area (**c**), and perimeter (**d**) of Actin-positive objects were normalized against the number of nuclei (Hoechst positive). Results are expressed as boxplots with individual measurements for each chip. Data were analyzed using independent T-test or Wilcoxon Rank-Sum Test (**** *p* ≤ 0.0001). Values were obtained from three independent experiments, with 4 to 5 chips per condition. (N = 3, n = 4–5).

**Figure 6 toxins-17-00089-f006:**
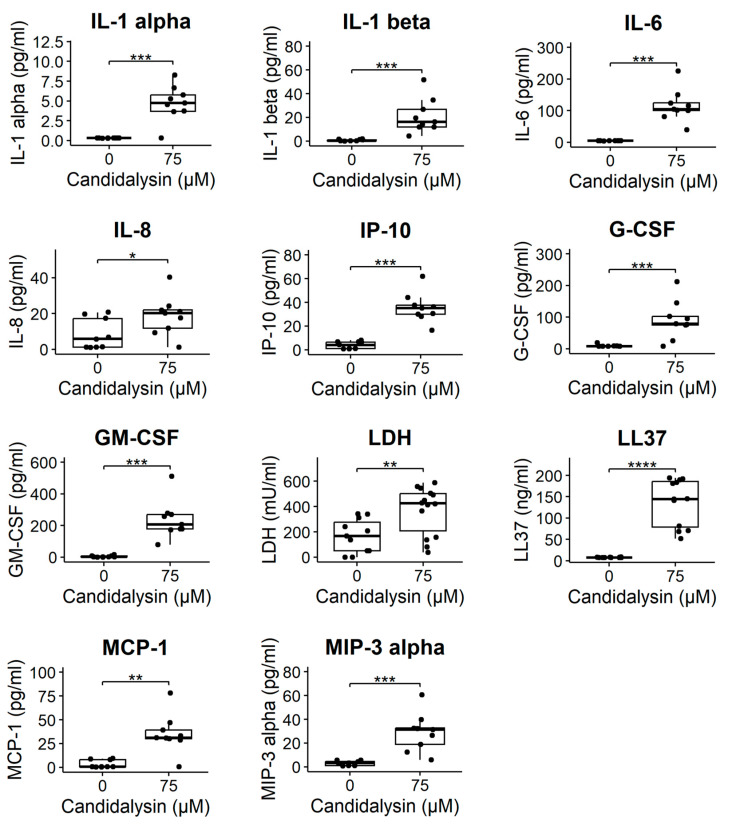
Candidalysin induces cytotoxicity and secretion of inflammatory markers in Caco-2 tubules. Candidalysin induced the secretion of MIP-3 alpha, G-CSF, GM-CSF, Il1-alpha, Il-1 beta, IL-6, IL-8, IP-10, MCP-1, LL37, and LDH in the apical side (tubule lumen). Results are expressed as boxplots with individual measurements for each chip. Data were analyzed using independent T-test or Wilcoxon Rank-Sum Test (* *p* ≤ 0.05; ** *p* ≤ 0.01; *** *p* ≤ 0.001, **** *p* ≤ 0.0001). Values were obtained from three independent experiments, with 3 to 5 chips per condition. (N = 3, n = 3–5).

**Figure 7 toxins-17-00089-f007:**
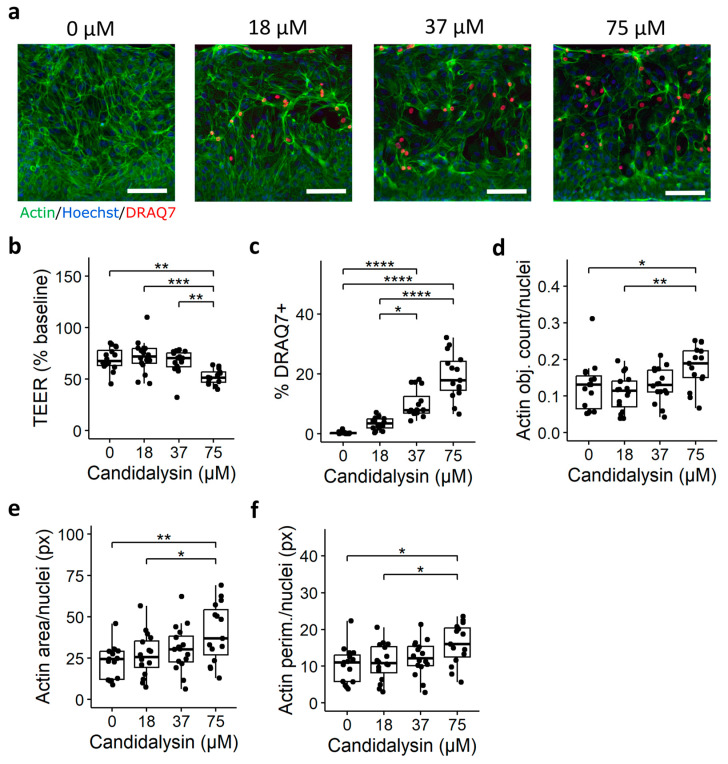
Candidalysin exposures in human colon organoid tubules. (**a**) colon organoid tubules exposed to 0, 18, 37, and 75 µM candidalysin for 120 min were stained with actin (green), DRAQ7 (red), and Hoechst (blue). Representative images are max projections taken at the bottom of the tubules. Scalebar is 100 µm. (**b**) TEER measurement after candidalysin exposure. Each chip is normalized to its value at baseline (before exposure). (**c**) The number (**d**), total area (**e**), and perimeter (**f**) of actin-positive objects were normalized against the number of nuclei (Hoechst positive). Results are expressed as boxplots with individual measurements for each chip. Data were analyzed using Kruskal–Wallis test with Dunn’s post hoc test or one-way ANOVA with Tukey’s HSD post hoc test (* *p* ≤ 0.05; ** *p* ≤ 0.01; *** *p* ≤ 0.001, **** *p* ≤ 0.0001, N = 3). Values were obtained from three independent experiments, with 4 to 6 chips per condition. (N = 3, n = 4–6).

## Data Availability

The raw data supporting the results of this article will be made available by the authors on request.

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
