# Peer review of "Breaking Barriers: Candidalysin Disrupts Epithelial Integrity and Induces Inflammation in a Gut-on-Chip Model"

_toxins, 2025, doi:10.3390/toxins17020089_

Round 1

Reviewer 1 Report

Comments and Suggestions for Authors

In this manuscript the authors use gut on a chip models to characterize the effects of candidalysin intoxication. This is a well conducted study and nicely written manuscript. The authors provide key proof of concept evidence for the relevance of using  gut on chip technology to assess Candida infections.

Major comments

1.     Is there a rational for selecting these two specific concentrations of candidalysin (37 and 75 uM)? Have the authors confirmed if these concentrations induce damage to the Caco2 tubules or are they inducing cell death?

2.     Having into account that TEER is reduced by 50% after candidalysin treatment, how do the authors justify such a low leakage level of the intestinal barrier? Although permeability to fluorescent dyes is significantly different between non-intoxicated and intoxicated conditions, these differences are in many instances less than 2-fold.

3.     Could the authors clarify how the CellProfiler pipeline was designed (Figure 5) to assess actin remodeling? How can the authors assign the actin clumps to the respective nuclei? It would be more appropriate to delimitate the entire cell and then quantify both the number of cells with actin clumps or the number of actin clumps per cell within the delimitated cells.

4.     The relevance of cytokine release can only truly be assessed if the assay detection limit is shown. The limit of detection of each cytokine assay could be shown either in the materials and methods section or presented as a dashed line in the respective plots (Figure 6). Also, the levels of S100A8/A9 were likely not detected, as these molecules are mainly released by neutrophils and monocytes. The authors might want to include this information.

Minor comments

1.     Figure 3a-c is not mentioned in the main text. Perhaps this legend could be explained in the main text after line 108.

2.     The authors use interchangeably the nomenclature “120min” and “2h”. Perhaps select one of these and keep consistency for clarity.

3.     The units in the y-axis of plots in Figure 5c and 5d are missing.

4.     Shouldn’t the TEER be normalized to 100% in the 0 uM condition in the plot of Figure 7a?

Reviewer 2 Report

Comments and Suggestions for Authors

Comments and suggestions are provided in the Review report.

Reviewer 3 Report

Comments and Suggestions for Authors

This paper describes the candidalysin disruption using gut-on-chip model.

This work provides an interesting approach using the OrganoPlate, but I have some questions and comments before the publication.

Major points

(1)

I do not know the OrganoPlate. So I have questions.

Usually, we monitor the TEER with the system with a membrane.

https://cellqart.com/applications/transepithelial-transendothelial-electrical-resistance-teer

Is this system a membrane-free system?

Are there any merits to use the system?

Moreover, please let me know the difference between Caco-2 tubules and Caco-2 cells.

Does the author use the Caco-2 tubules in OrganoPlate, not Caco-2 cells?

As shown in the website, there should be two separation phases. However, OrganoPlate has three separation channels. Red(cells) and blue(buffer) are enough to separate. Why are there three separation channels? In Fig2b, are red(cells)and blue(buffer or candidalysin solution) enough to measure the disruption?

(2)

Fig3d

The author concludes that sodium fluorescein was significantly increased after 30-, 60-, and 120- incubation with 37uM candidalysin. On the other hand, Dextran significantly increased after 30 min 37uM candidalysin.

75uM candidalysin increased permeability to both dyes after all incubation times.

I could not see any increase in these results.

In the discussion(p10,L223), other groups did not observe a decrease in TEER.

I wondered about the author's interpretation.

Minor points

(1)

Please describe the candidalysin sequence and MW.

(2)

scale bar 100uM> 100um?

Round 2

Reviewer 2 Report

Comments and Suggestions for Authors

The authors addressed all the comments and the comments are satisfactory. The paper is acceptable at it's current form. 

Reviewer 3 Report

Comments and Suggestions for Authors

All of my concerns were addressed properly.